# Data Sharing and Compression for Cooperative Networked Control

**Jiangnan Cheng**[1], **Marco Pavone**[2], **Sachin Katti**[3], **Sandeep Chinchali**[4], **and Ao Tang**[1]

[1]School of Electrical and Computer Engineering, Cornell University, Ithaca, NY
[2]Department of Aeronautics and Astronautics, Stanford University, Stanford, CA
[3]Department of Computer Science, Stanford University, Stanford, CA
[4]Department of Electrical and Computer Engineering, The University of Texas at Austin, Austin, TX
{jc3377, atang}@cornell.edu, {pavone, skatti}@stanford.edu,
sandeepc@utexas.edu

## Abstract

Sharing forecasts of network timeseries data, such as cellular or electricity load patterns, can improve independent control applications ranging from traffic scheduling to power generation. Typically, forecasts are designed without knowledge of a downstream controller's task objective, and thus simply optimize for *mean* prediction error. However, such task-agnostic representations are often too large to stream over a communication network and do not emphasize salient temporal features for cooperative control. This paper presents a solution to learn succinct, highly-compressed forecasts that are *co-designed* with a modular controller's task objective. Our simulations with real cellular, Internet-of-Things (IoT), and electricity load data show we can improve a model predictive controller's performance by at least $25\%$ while transmitting $80\%$ less data than the competing method. Further, we present theoretical compression results for a networked variant of the classical linear quadratic regulator (LQR) control problem.

## 1 Introduction

Cellular network and power grid operators measure rich timeseries data, such as city-wide mobility and electricity demand patterns. Sharing such data with *external* entities, such as a taxi fleet operator, can enhance a host of societal-scale control tasks, ranging from taxi routing to battery storage optimization. However, how should timeseries owners *represent* their data to limit the scope and volume of information shared across a data boundary, such as a congested wireless network?[1]

At a first glance, it might seem sufficient to simply share generic demand forecasts with any downstream controller. Each controller, however, often has a unique cost function and context-specific sensitivity to prediction errors. For example, cell demand forecasts should emphasize accurate peak-hour forecasts for taxi fleet routing. The same underlying cellular data should instead emphasize fine-grained throughput forecasts when a video streaming controller starts a download. Despite the benefits of customizing forecasts for control, today's forecasts are mostly *task-agnostic* and simply optimize for mean or median prediction error. As such, they often waste valuable network bandwidth to transmit temporal features that are unnecessary for a downstream controller. Even worse, they might not minimize errors when they matter most, such as peak-hour variability.

Given the limitations of today's task-agnostic forecasts, this paper contributes a novel problem formulation for learning *task-driven* forecasts for networked control. In our general problem (Fig.

---

[1]Uber processes petabytes of data per day [1] and a mobile operator can process 60 TB of daily cell metrics [2]. Even a *fraction* of such data is hard to send.

35th Conference on Neural Information Processing Systems (NeurIPS 2021).

1), an operator measures timeseries $s_t$, such as electricity or cell demand, and transmits compressed representation $\phi_t$, which is decoded to $\hat{s}_t$ at the controller. Rather than simply minimize the prediction error for $\hat{s}_t$, we instead learn a representation that minimizes a modular controller $\pi$'s ultimate cost $J$. Our key technical insight is to compute a controller's sensitivity to prediction errors, which in turn guides how we *co-design* and learn a concise forecast representation that is tailored to control. As such, our scheme jointly integrates data-driven forecasting, compression, and model-based control.

**Related work:** Our work is broadly related to information-theoretic compression for control as well as task-driven representation learning. The closest work to ours is [3], where task-driven forecasts are learned for one-step stochastic optimization problems. In stark contrast, we address *compression* of timeseries forecasts and focus on networked, multi-step control problems. Our work is also inspired by Shannon's rate-distortion theory [4], which describes how to encode and transmit signals with a minimal bit-rate to minimize reconstruction error. In contrast, we work with real numbers rather than bits and focus on reducing the dimension of data while keeping task-specific control cost low.

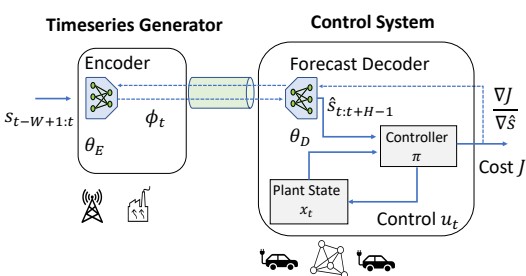

Figure 1: **Data sharing for cooperative control:** An owner of timeseries data $s_t$, such as a mobile operator, needs to transmit a compressed representation $\phi_t$ to a downstream controller with internal state $x_t$. The *learned* forecast emphasizes task-relevant temporal features to minimize end-to-end controller cost $J$.

Prior work has addressed rate-distortion trade-offs for networked LQR control problems [5–7]. However, these works focus on ensuring closed-loop stability for a remote controller and a physically-separated plant, such as in tele-operation. Our problem is fundamentally different, since we address how *external* timeseries forecasts can enhance a controller's *local* decisions using full knowledge of its own internal state. While the term *co-design* appears in select work on networked LQR, it refers to a drastically different setting where a communication scheduler and tele-operated controller must be jointly designed [8–11]. Moreover, event-triggered control/learning [12–15] emphases temporal sparsity of communications, while in our setting the MPC controller consistently requires a forecast of timeseries. Finally, our work differs from deep neural network (DNN) compression schemes for video inference [16, 17] since we focus on control.

More discussions will follow in Sec. 2 after the technical problem is introduced in detail.

**Contributions:** In light of prior work, our contributions are three-fold. First, we introduce a novel problem for learning compressed timeseries representations that are tailored to control. Second, to gain insights into our problem, we contribute analytic compression results for LQR control. These insights serve as a foundation for our general algorithm that computes the sensitivity of a model predictive controller (MPC) to prediction errors, which guides learning of concise forecast representations. Third, we learn representations that improve control performance by $> 25\%$ and are $80\%$ smaller than those generated by standard autoencoders, even for real IoT sensor data we captured on embedded devices as well as benchmark electricity and cell datasets.

**Organization:** In Sec. 2, we formalize a general problem of compression for networked control and provide analytical results for LQR. Then, in Sec. 3, we contribute an algorithm for task-driven data compression for general MPC problems. We demonstrate strong empirical performance of our algorithm for cell, energy, and IoT applications in Sec. 4 - 5 and conclude in Sec. 6.

## 2   Problem Formulation

We now describe the information exchange between a generator of timeseries data, henceforth called a forecaster, and a controller, as shown in Fig. 1. Both systems operate in discrete time, indexed by $t$, for a time horizon of $T$ steps. The notation $y_{a:b}$ denotes a timeseries $y$ from time $a$ to $b$.

**Forecast Encoder:** The forecaster measures a high-volume timeseries $s_t \in \mathbb{R}^p$. Timeseries $s$ is drawn from a domain-specific distribution $\mathcal{D}$, such as cell-demand patterns, denoted by $s_{0:T-1} \sim \mathcal{D}$.

A differentiable encoder maps the past $W$ measurements, denoted by $s_{t-W+1:t}$, to a compressed representation $\phi_t \in \mathbb{R}^Z$, using model parameters $\theta_e$: $\phi_t = g_{\text{encode}}(s_{t-W+1:t}; \theta_e)$. Typically, $Z \ll p$ and is referred to as the *bottleneck* dimension since it limits the communication data-rate and how many floating-point values are sent per unit time.

**Forecast Decoder:** The compressed representation $\phi_t$ is transmitted over a bandwidth-constrained communication network, where a downstream decoder maps $\phi_t$ to a forecast $\hat{s}_{t:t+H-1}$ for the next $H$ steps, denoted by: $\hat{s}_{t:t+H-1} = g_{\text{decode}}(\phi_t; \theta_d)$, where $\theta_d$ are decoder parameters. Importantly, we decode representation $\phi_t$ into a forecast $\hat{s}$ so it can be directly passed to a model-predictive controller that interprets $\hat{s}$ as a physical quantity, such as traffic demand. The encoder and decoder jointly enable compression and forecasting by mapping past observations to a forecast via bottleneck $\phi_t$.

**Modular Controller:** The controller has an internal state $x_t \in \mathbb{R}^n$ and must choose an optimal control $u_t \in \mathbb{R}^m$. We denote the admissible state and control sets by $\mathcal{X}$ and $\mathcal{U}$ respectively. The system dynamics also depend on external timeseries $s_t$ and are given by: $x_{t+1} = f(x_t, u_t, s_t)$, $t \in \{0, \cdots, T-1\}$. Importantly, while state $x_t$ depends on *exogenous* input $s_t$, we assume $s_t$ evolves *independently* of $x_t$ and $u_t$. This is a practical assumption in many networked settings. For example, the demand $s_t$ for taxis might mostly depend on city commute patterns and not an operator's routing decisions $u_t$ or fleet state $x_t$. Ideally, control policy $\pi$ chooses a decision $u_t$ based on fully-observed internal state $x_t$ and *perfect* knowledge of exogenous input $s_{t:t+H-1}$: $u_t = \pi(x_t, s_{t:t+H-1}; \theta_c)$, where $\theta_c$ are control policy parameters, such as a feedback matrix for LQR. However, in practice, given a possibly noisy forecast $\hat{s}_{t:t+H-1}$, it will *enact* a control denoted by $\hat{u}_t = \pi(x_t, \hat{s}_{t:t+H-1}; \theta_c)$, which implicitly depends on the encoder/decoder parameters $\theta_e, \theta_d$ via the forecast $\hat{s}$.

**Control Cost:** Our main objective is to minimize end-to-end control cost $J^c$, which depends on initial state $x_0$ and controls $\hat{u}_{0:T-1}$, which in turn depend on the *forecast* $\hat{s}_{0:T-1}$. For a simpler notation, we use bold variables to define the full timeseries, i.e., $\mathbf{u} := u_{0:T-1}$, $\mathbf{s} := s_{0:T-1}$, $\hat{\mathbf{u}} := \hat{u}_{0:T-1}$ and $\hat{\mathbf{s}} := \hat{s}_{0:T-1}$. The control cost $J^c$ is a sum of stage costs $c(x_t, \hat{u}_t)$ and terminal cost $c_T(x_T)$: $J^c(\hat{\mathbf{u}}; x_0, \mathbf{s}) = c_T(x_T) + \sum_{t=0}^{T-1} c(x_t, \hat{u}_t)$, where $x_{t+1} = f(x_t, \hat{u}_t, s_t), t \in \{0, \cdots, T-1\}$. Importantly, the above plant dynamics $f$ evolve according to true timeseries $s_t$, but controls $\hat{u}_t$ are enacted with possibly noisy forecasts $\hat{s}_t$.

**Forecasting Errors:** In practice, a designer often wants to visualize decoded forecasts $\hat{s}$ to debug anomalies or view trends. While our principal goal is to minimize the control errors and cost associated with forecast $\hat{s}$, we allow a designer to *optionally* penalize mean squared prediction error (MSE). This penalty incentivizes a forecast $\hat{s}_t$ to estimate the key trends of $s_t$, serving as a regularization term: $J^F(\mathbf{s}, \hat{\mathbf{s}}) = \frac{1}{T} \sum_{t=0}^{T-1} ||s_t - \hat{s}_t||_2^2$.

**Overall Weighted Cost:** Given our principal objective of minimizing control cost and optionally penalizing prediction error, we combine the two costs using a user-specified weight $\lambda^F$. Importantly, we try to minimize the *additional* control cost $J^c(\hat{\mathbf{u}}; x_0, \mathbf{s})$ incurred by using forecast $\hat{\mathbf{s}}$ instead of true timeseries $\mathbf{s}$, yielding overall cost:

$$J^{\text{tot.}}(\mathbf{u}, \hat{\mathbf{u}}, \mathbf{s}, \hat{\mathbf{s}}; x_0, \lambda^F) = \frac{1}{T}\big( \underbrace{J^c(\hat{\mathbf{u}}; x_0, \mathbf{s}) - J^c(\mathbf{u}; x_0, \mathbf{s})}_{\text{extra control cost}} \big) + \lambda^F J^F(\mathbf{s}, \hat{\mathbf{s}}). \tag{1}$$

The total cost implicitly depends on controller, encoder, and decoder parameters via controls $\mathbf{u}$ and $\hat{\mathbf{u}}$ and the forecast $\hat{\mathbf{s}}$. Having defined the encoder/decoder and controller, we now formally define the problem addressed in this paper.

**Problem 1** (Data Compression for Cooperative Networked Control). *We are given a controller $\pi(\cdot; \theta_c)$ with fixed, pre-trained parameters $\theta_c$, fixed bottleneck dimension $Z$, and perfect measurements of internal controller state $x_{0:T}$. Given a true exogenous timeseries $s_{0:T-1}$ drawn from data distribution $\mathcal{D}$, find encoder and decoder parameters $\theta_e, \theta_d$ to minimize the weighted control and forecasting cost (Eq. 1) with weight $\lambda^F$:*

$$\theta_e^*, \theta_d^* = \underset{\theta_e, \theta_d}{\arg\min} \quad \mathbb{E}_{s_{0:T-1} \sim \mathcal{D}}[J^{\text{tot.}}(\mathbf{u}, \hat{\mathbf{u}}, \mathbf{s}, \hat{\mathbf{s}}; x_0, \lambda^F)], \text{ where}$$

$$\phi_t = g_{\text{encode}}(s_{t-W+1:t}; \theta_e), \ \phi_t \in \mathbb{R}^Z$$
$$\hat{s}_{t:t+H-1} = g_{\text{decode}}(\phi_t; \theta_d),$$
$$\hat{u}_t = \pi(x_t, \hat{s}_{t:t+H-1}; \theta_c), \ u_t = \pi(x_t, s_{t:t+H-1}; \theta_c),$$
$$x_{t+1} = f(x_t, \hat{u}_t, s_t), \quad \text{and} \quad x_t \in \mathcal{X}, \hat{u}_t \in \mathcal{U}, \quad t \in \{0, \cdots, T-1\}.$$

**Technical Novelty and Practicality of our Co-design Problem:**

Having formalized our problem, we can now articulate how it differs from classical networked control and tele-operation [18–20, 6, 5], compressed sensing [21, 22], and certainty-equivalent control [23, 24]. First, we can not readily apply the classical separation principle [25] of Linear Quadratic Gaussian (LQG) control, which proscribes how to *independently* design a timeseries estimator, such as the Kalman Filter [26], and a "certainty-equivalent" controller (the linear quadratic regulator) for optimal performance. This is because the timeseries owner measures a **non-stationary timeseries** $s_t$ (e.g. spatiotemporal cell demand patterns), without an analytical process model for standard Kalman Filtering, motivating our subsequent use of learned DNN forecasters. Second, due to **data-rate constraints**, we must prioritize task-relevant features as opposed to equally weighting and sending the full $\hat{s}_t$, which a classic state observer in LQG would do.

Moreover, even when the estimator and controller are separated by a bandwidth-limited network and the separation principle does not hold [27], our setting still differs from classical networked control [18–20, 6, 5]. These works assume that both the full plant state $x_t$ and controls $u_t$ are encoded and transmitted between a remote controller and plant. In stark contrast, the only transmitted data in our setting is *external* information $s_t$ from a network operator, which can improve an independent controller's *local* decisions $u_t$ based on its internal state $x_t$. As such, simply grouping controller state $x_t$ and network timeseries $s_t$ into a joint state for classical tele-operation is infeasible, since $x_t$ and $s_t$ are measured at different locations by different entities. In essence, Prob. 1 formalizes how a network operator can provide significant value to an independent controller by judicious data sharing.

## 3  Forecaster and Controller Co-design

Prob. 1 is of wide scope, and can encompass both neural network forecasters and controllers. For intuition, we first provide analytical results for an *input-driven* LQR problem in Sec. 3.1. We then use such insights in a general learning algorithm that scales to DNN forecasters in Sec. 3.2.

### 3.1  Input-Driven Linear Quadratic Regulator (LQR)

We first consider a simple instantiation of Prob. 1 with linear dynamics[2], no state or control constraints, and a quadratic control cost. Since the dynamics have linear dependence on the exogenous input $s$, we refer to this setting as an *input-driven* LQR problem. We first analyze the problem when controls are computed for the full-horizon from $t = 0$ to $T = H$ and then extend to receding-horizon control (MPC) in Sec. 3.2. The dynamics and control cost are:

$$x_{t+1} = Ax_t + Bu_t + Cs_t, \tag{2}$$

$$J^c = \sum_{t=0}^{H} x_t^\top Q x_t + \sum_{t=0}^{H-1} u_t^\top R u_t, \tag{3}$$

where $Q, R$ are positive definite. Our first step is to determine the optimal control. Given the linear dynamics, for all times $i \in \{0, \cdots, H-1\}$, each $x_{i+1}$ is a linear function of initial condition $x_0$ and the *full future* control vector $\mathbf{u}$ and $\mathbf{s}$:

$$x_{i+1} = A^{i+1} x_0 + \boldsymbol{M_i} \mathbf{u} + \boldsymbol{N_i} \mathbf{s}, \quad \text{where} \tag{4}$$

$$\boldsymbol{M_i} = \begin{bmatrix} A^i B & A^{i-1}B & \cdots & B & \mathbf{0} \end{bmatrix} \in \mathbb{R}^{n \times mH}, \boldsymbol{N_i} = \begin{bmatrix} A^i C & A^{i-1}C & \cdots & C & \mathbf{0} \end{bmatrix} \in \mathbb{R}^{n \times pH}.$$

Therefore, given $x_0$ and vector $\mathbf{s}$, control cost $J^c$ is a quadratic function of $\mathbf{u}$:

$$J^c(\mathbf{u}; x_0, \mathbf{s}) \;=\; \mathbf{u}^\top \underbrace{(\boldsymbol{R} + \sum_{i=0}^{H-1} \boldsymbol{M_i}^\top Q \boldsymbol{M_i})}_{\boldsymbol{K}} \mathbf{u} + 2 \underbrace{[\sum_{i=0}^{H-1} \boldsymbol{M_i}^\top Q (A^{i+1}x_0 + \boldsymbol{N_i}\mathbf{s})]^\top}_{\boldsymbol{k}(x_0, \mathbf{s})} \mathbf{u} + \text{ constant},$$

$$\tag{5}$$

---

[2]Transition noise is not added here due to certainty equivalence of input-driven LQR, as shown in Appendix Sec. A.2.

where the constant of $\sum_{i=0}^{H-1}(A^{i+1}x_0 + \boldsymbol{N_i}\mathbf{s})^\top Q(A^{i+1}x_0 + \boldsymbol{N_i}\mathbf{s})$ is independent of $\mathbf{u}$, and $\boldsymbol{R} = \text{blockdiag}(R, \cdots, R) \in \mathbb{R}^{mH \times mH}$. Clearly, $\boldsymbol{K}$ is positive definite and $J^c$ is strictly convex. Given the convex quadratic cost, the optimal control is $\mathbf{u}^*$, where $\mathbf{u}^* = -\boldsymbol{K}^{-1}\boldsymbol{k}(x_0, \mathbf{s})$. However, given a possibly noisy forecast $\hat{\mathbf{s}}$, we would instead plan and enact controls denoted by $\hat{\mathbf{u}}$, where $\hat{\mathbf{u}} = -\boldsymbol{K}^{-1}\boldsymbol{k}(x_0, \hat{\mathbf{s}})$. Thus, the sensitivity of such controls to forecast errors is:

$$\hat{\mathbf{u}} - \mathbf{u}^* = -\boldsymbol{K}^{-1}(\boldsymbol{k}(x_0, \hat{\mathbf{s}}) - \boldsymbol{k}(x_0, \mathbf{s})) = -\boldsymbol{K}^{-1}\underbrace{(\sum_{i=0}^{H-1}\boldsymbol{M_i}^\top Q\boldsymbol{N_i})}_{\boldsymbol{L}}(\hat{\mathbf{s}} - \mathbf{s}), \quad (6)$$

and the sensitivity of the control cost to forecast errors is:

$$J^c(\hat{\mathbf{u}}; x_0, \mathbf{s}) - J^c(\mathbf{u}^*; x_0, \mathbf{s}) = (\hat{\mathbf{u}} - \mathbf{u}^*)^\top \boldsymbol{K}(\hat{\mathbf{u}} - \mathbf{u}^*) = (\hat{\mathbf{s}} - \mathbf{s})^\top \underbrace{\boldsymbol{L}^\top \boldsymbol{K}^{-1}\boldsymbol{L}}_{\text{co-design matrix } \Psi}(\hat{\mathbf{s}} - \mathbf{s}), \quad (7)$$

where we term the positive semi-definite *co-design matrix* $\Psi = \boldsymbol{L}^\top \boldsymbol{K}^{-1}\boldsymbol{L}$. We now combine the extra control cost and prediction error to calculate the total cost as:

$$J^{\text{tot.}} = \frac{1}{H}\Big(\underbrace{(\hat{\mathbf{s}} - \mathbf{s})^\top \Psi(\hat{\mathbf{s}} - \mathbf{s})}_{\text{extra control cost}} + \lambda^{\text{F}}\underbrace{(\hat{\mathbf{s}} - \mathbf{s})^\top (\hat{\mathbf{s}} - \mathbf{s})}_{\text{prediction error}}\Big) = \frac{1}{H}\big((\hat{\mathbf{s}} - \mathbf{s})^\top (\Psi + \lambda^{\text{F}}I)(\hat{\mathbf{s}} - \mathbf{s})\big). \quad (8)$$

The above expression leads to an intuitive understanding of co-design. The co-design matrix $\Psi$ in Eq. 7 essentially weights the error in elements of $\hat{\mathbf{s}}$ based on their importance to the ultimate control cost. Thus, our approach is fundamentally *task-aware* since the co-design matrix depends on LQR's dynamics, control, and cost matrices as shown in Eq. 6 and 7. The optional weighting of prediction error with $\lambda^{\text{F}}$ acts as a regularization term. Moreover, we now show that we can reduce input-driven LQR to a low-rank approximation problem, which allows us to find an *analytic* expression for an optimal encoder/decoder.

**Input-Driven LQR is Low-Rank Approximation.** Given the above expressions for the total cost, we now assume a simple parametric model for the encoder and decoder to formally write Prob. 1 for the toy input-driven LQR setting. Specifically, we assume a linear encoder $E \in \mathbb{R}^{Z \times pH}$ maps true exogenous input $\mathbf{s}$ to representation $\phi = E\mathbf{s}$, where $\phi \in \mathbb{R}^Z$. Then, linear decoder matrix $D \in \mathbb{R}^{pH \times Z}$ yields decoded timeseries $\hat{\mathbf{s}} = D\phi = DE\mathbf{s}$. In practice, we often have a training dataset consisting of $N$ samples of exogenous input $\mathbf{s}$ drawn from a data distribution $\mathbf{s} \sim \mathcal{D}$. These samples can be arranged as columns in a matrix $\mathbf{S} \in \mathbb{R}^{pH \times N}$. To learn an encoder $E$ and decoder $D$ from $N$ samples $\mathbf{S}$ at once, we can express our problem as:

$$\underset{D,E}{\text{argmin}} \quad \sum_{i=1}^{N}(\hat{\mathbf{S}}_i - \mathbf{S}_i)^\top (\Psi + \lambda^{\text{F}}I)(\hat{\mathbf{S}}_i - \mathbf{S}_i), \quad \text{where}$$
$$\hat{\mathbf{S}} = DE\mathbf{S}, \text{ rank}(D) \le Z \text{ and rank}(E) \le Z, \quad (9)$$

where $\mathbf{S}_i$ and $\hat{\mathbf{S}}_i$ represent the $i$-th column vector of $\mathbf{S}$ and $\hat{\mathbf{S}}$. We now characterize the input-driven LQR problem.

**Proposition 1** (Linear Weighted Compression). *Input-driven LQR (Eq. 9) is a low-rank approximation problem, which admits an analytical solution for an optimal encoder and decoder pair $(E, D)$.*

*Proof.* We first re-write the objective of the input-driven LQR problem (Eq. 9) as: $\sum_{i=1}^{N}(\hat{\mathbf{S}}_i - \mathbf{S}_i)^\top (Y\Lambda Y^\top)(\hat{\mathbf{S}}_i - \mathbf{S}_i) = ||\Lambda^{\frac{1}{2}}Y^\top\hat{\mathbf{S}} - \Lambda^{\frac{1}{2}}Y^\top\mathbf{S}||_F^2$, where $Y\Lambda Y^\top$ is the eigen-decomposition of the positive definite matrix $\Psi + \lambda^{\text{F}}I$ and $||.||_F$ represents the Frobenius norm of a matrix. Thus, the problem can be written as:

$$\underset{D,E}{\text{argmin}} \quad ||\underbrace{\Lambda^{\frac{1}{2}}Y^\top DE\mathbf{S}}_{\text{approximation}} - \underbrace{\Lambda^{\frac{1}{2}}Y^\top\mathbf{S}}_{\text{original}}||_F^2, \quad \text{where } \text{rank}(D) \le Z \text{ and rank}(E) \le Z, \quad (10)$$

which is the canonical form of a low-rank approximation problem. By the Eckhart-Young theorem, the solution to the input-driven LQR problem (Eq. 10) is the rank $Z$ truncated singular value decomposition (SVD) of original matrix $\Lambda^{\frac{1}{2}}Y^\top\mathbf{S}$, denoted by $U\Sigma V^\top$. In the truncated SVD,

$U \in \mathbb{R}^{pH \times Z}$ is semi-orthogonal, $\Sigma \in \mathbb{R}^{Z \times Z}$ is a diagonal matrix of singular values, and $V \in \mathbb{R}^{N \times Z}$ is semi-orthogonal. Further, an encoder of $E = U^\top \Lambda^{\frac{1}{2}} Y^\top$ and decoder of $D = (\Lambda^{\frac{1}{2}} Y^\top)^{-1} U$ solve the problem since:

$$\underbrace{\Lambda^{\frac{1}{2}} Y^\top D E \mathbf{S}}_{\text{approximation}} = \Lambda^{\frac{1}{2}} Y^\top \underbrace{(\Lambda^{\frac{1}{2}} Y^\top)^{-1} U}_{D} \underbrace{U^\top \Lambda^{\frac{1}{2}} Y^\top}_{E} \mathbf{S} = U(U^\top \Lambda^{\frac{1}{2}} Y^\top \mathbf{S}) = \underbrace{U \Sigma V^\top}_{\text{optimal rank Z approximation}} .$$

A similar analysis for a linear encoder-decoder structure for networked inference, not control, is presented in [17]. The key difference from our current paper is our problem setup is for control, not networked inference. Moreover, our result differs from existing LQR literatures with exogenous input, such as [28], since our exogenous input $s_t$ is subject to to a network bottleneck and encoder/decoder, which is the crux of our Prob. 1; and our total cost includes the extra control cost due to mis-estimation of $s_t$, rather than simply the prediction error of $s_t$.

**Compression benefits:** Casting input-driven LQR as low-rank approximation provides significant intuition. As shown in Proposition 1, the optimal encoder/decoder depend on the truncated SVD of $\Lambda^{\frac{1}{2}} Y^\top \mathbf{S}$, which takes into account the control task via the co-design matrix, importance of prediction errors via $\lambda^{\mathrm{F}}$, and statistics of the input via $\mathbf{S}$. We achieved strong compression benefits for simulations of input-driven LQR (provided in supplement Fig. 5 due to space limits).

**Transitioning to Model Predictive Control (MPC).** In practice, we often have forecasts for a short horizon $H < T$. Then, starting from any state $x_t$, MPC will plan a sequence of controls $\hat{u}_{t:t+H-1}$, enact the first control $\hat{u}_t$, and then re-plan with the next forecast. If we replace the horizon to $H < T$ in the input-driven LQR analysis in Sec. 3.1, $\mathbf{u}^* = -\boldsymbol{K}^{-1} \boldsymbol{k}(x_0, \mathbf{s})$ gives the optimal control for a *short-horizon $H$* and we can encode/decode using a low rank approximation as in Prop. 1. While the performance is *not* necessarily optimal for the full duration $T$, MPC performs extremely well in practice, yielding even better compression gains, as shown in the supplement (Fig. 6).

---

**Algorithm 1** Compression Co-design for Control

1: Set forecast weight $\lambda^{\mathrm{F}}$, bottleneck size $Z$
2: Initialize encoder/decoder parameters $\theta_{\mathrm{e}}^0, \theta_{\mathrm{d}}^0$ randomly, and fix controller parameters $\theta_{\mathrm{c}}$
3: **for** $\tau \leftarrow 0$ to $N_{\mathrm{epoch}} - 1$ **do**
4:     Initialize Controller State $x_0 \in \mathcal{X}$
5:     **for** $t \leftarrow 0$ to $T - 1$ **do**
6:         Encode $\phi_t = g_{\mathrm{encode}}(s_{t-W+1:t}; \theta_{\mathrm{e}}^\tau)$
7:         Decode $\hat{s}_{t:t+H-1} = g_{\mathrm{decode}}(\phi_t; \theta_{\mathrm{d}}^\tau)$
8:         Enact $\hat{u}_t = \pi(x_t, \hat{s}_{t:t+H-1}; \theta_{\mathrm{c}})$
9:         Propagate $x_{t+1} \leftarrow f(x_t, \hat{u}_t, s_t)$
10:        $u_t = \pi(x_t, s_{t:t+H-1}; \theta_{\mathrm{c}})$ (For Training Only)
11:     **end for**
12:     $\theta_{\mathrm{e}}^{\tau+1}, \theta_{\mathrm{d}}^{\tau+1} \leftarrow$
        $\textsc{BackProp}\big[J^{\mathrm{tot}\cdot}(\mathbf{u}, \hat{\mathbf{u}}, \mathbf{s}, \hat{\mathbf{s}}; x_0, \lambda^{\mathrm{F}})\big]$
13: **end for**
14: Return learned parameters $\theta_{\mathrm{e}}^{N_{\mathrm{epoch}}}, \theta_{\mathrm{d}}^{N_{\mathrm{epoch}}}$

---

We also note a practitioner can adopt a simple cost function based on MPC that complements Eq. 1. The MPC controller $\pi$ will optimize the cost $J^{\mathrm{tot}\cdot}$ given a short-horizon forecast $\hat{s}_{t:t+H-1}$, but only enact the *first* control $\hat{u}_t = \pi(x_t, \hat{s}_{t:t+H-1}; \theta_{\mathrm{c}})$. Meanwhile, the best first control MPC can take is $u_t = \pi(x_t, s_{t:t+H-1}; \theta_{\mathrm{c}})$ with perfect knowledge of $s$ for horizon $H$. Thus, our insight is that we can penalize the errors in *enacted* controls $\hat{u}_t$ during training and regularize for prediction error, using cost: $\frac{1}{T}\big(\sum_{t=0}^{T-1} ||\hat{u}_t - u_t||_2^2 + \lambda^{\mathrm{F}} ||\hat{s}_t - s_t||_2^2\big)$. In our experiments, we observed strong performance by optimizing for the cost Eq. 1, as well as the above cost, which optimizes $J^{\mathrm{tot}\cdot}$ over a short-horizon for MPC. We now crystallize these insights from input-driven LQR into a formal algorithm that applies to data-driven MPC.

## 3.2 Algorithm to Co-design Forecaster and Controller

For more complex scenarios than LQR, it is challenging to provide analytical forms of an optimal encoder and decoder. Thus, we present a heuristic algorithm to solve Prob. 1 in Algorithm 1. Our key technical insight is that, if the encoder, decoder, and controller are differentiable, we can write:

$$\frac{\nabla J^{\mathrm{tot}\cdot}(\mathbf{u}, \hat{\mathbf{u}}, \mathbf{s}, \hat{\mathbf{s}}; x_0, \lambda^{\mathrm{F}})}{\nabla \theta_{\mathrm{e}}} = \frac{\nabla J^{\mathrm{tot}\cdot}(\mathbf{u}, \hat{\mathbf{u}}, \mathbf{s}, \hat{\mathbf{s}}; x_0, \lambda^{\mathrm{F}})}{\nabla(\hat{\mathbf{s}} - \mathbf{s})} \times \frac{\nabla(\hat{\mathbf{s}} - \mathbf{s})}{\nabla \theta_{\mathrm{e}}}, \tag{11}$$

and likewise for $\theta_\mathrm{d}$. The first term captures the sensitivity of the control cost with respect to prediction errors and the second propagates that sensitivity to the forecasting model. Crucially, the gradient of $J^\mathrm{tot.}$ can be obtained from recent methods that learn differentiable MPC controllers [29, 30].

In lines 1-2 of Alg.1, we randomly initialize the encoder and decoder parameters and set the latent representation size $Z$ to limit the communication data-rate. Then, we enact control policy rollouts in lines 3-11 for $N_\mathrm{epoch}$ training epochs, each of duration $T$. We first encode and decode the forecast $\hat{\mathbf{s}}$ (lines 6-7) and pass them to the downstream controller with fixed parameters $\theta_\mathrm{c}$ (lines 8-10). During training, we calculate the loss by comparing the optimal weighted cost with *true* input $\mathbf{s}$ and the forecast $\hat{\mathbf{s}}$. In turn, this loss is used to train the differentiable encoder and decoder through backpropagation in line 12. Finally, the learned encoder and decoder (line 14) are deployed.

**Co-design Algorithm Discussion:** A few comments are in order. First, true input $\mathbf{s}$ is only needed during *training*, which is accomplished at a single server using historical data to avoid passing large gradients over a real network. Then, we can periodically re-train the encoder/decoder during online deployment. Second, our approach also applies when $\theta_\mathrm{c}$ are parameters of a deep reinforcement learning (RL) policy. However, since the networked systems we consider have well-defined dynamical models, we focus our evaluation on model-based control.

## 4 Application Scenarios

We now describe three diverse application scenarios addressed in our evaluation. The scenarios are linear MPC problems with box control constraints:

$$x_{t+1} = x_t + u_t - s_t, \quad \text{(Dynamics)} \quad \text{where} \quad u_\mathrm{min} \le u_t \le u_\mathrm{max}. \quad \text{(Constraints)} \quad (12)$$

Our scenarios have the same state and control dimensions $m = n$, and dynamics/control matrices $A = B = I_{n \times n}$ indicate uniform coupling between controls and the next state. Finally, we have actuation limits $u_\mathrm{min}$ and $u_\mathrm{max}$. The cost function incentivizes regulation of the state $x_t$ to a set-point $L$. In practice, we often want to penalize states below the set-point, such as inventory shortages where $x_t < L$, more heavily than those above, such as excesses. In the following cost, weights $\gamma_e, \gamma_s, \gamma_u \in \mathbb{R}^+$ govern excesses, shortages, and controls $u_t$ respectively:

$$J^\mathrm{c}(\mathbf{x}, \mathbf{u}) = \sum_{t=0}^{T} (\gamma_e ||[x_t - L]_+||_2^2 + \gamma_s ||[L - x_t]_+||_2^2) + \sum_{t=0}^{T-1} \gamma_u ||u_t||_2^2, \quad (13)$$

where $[x]_+$ represents the positive elements of a vector. We focus on linear MPC with box constraints and a flexible quadratic cost (Eq. 13) since it is a canonical problem [31, 32] with wide applications in networked systems. However, to show the generality of co-design, we provide strong experimental results for a mobile video streaming application with *noisy, non-linear* dynamics in Sec. 5.2. We evaluate diverse MPC settings coupled with an array of neural network forecasters.

**Smart Factory Regulation with IoT Sensors:** We consider an *idealized* scenario similar to datacenter temperature control [33], where $x_t \in \mathbb{R}^n$ represents the temperature, humidity, pressure and light for $\frac{n}{4}$ machines in a smart factory, each of whose 4 sensor measurements we want to regulate to a set-point of $L$. External heat, humidity, and pressure disturbances $s \in \mathbb{R}^p$ add to state $x_t$ in the dynamics (Eq. 12). Disturbances are measured by $p = n$ IoT sensors, such as from nearby heating units. Our objective is to select control inputs $u \in \mathbb{R}^m$ to regulate the environment *anticipating* disturbances $s$ from the $p$ IoT sensors. The cost function (Eq. 13) has $\gamma_e = \gamma_s = \gamma_u = 1$ to equally penalize deviation from the set-point and regulation effort. Finally, we collected two weeks of stochastic timeseries of temperature, pressure, humidity, and light from the Google Edge Tensor Processing Unit (TPU)'s environmental sensor board for our experiments, as detailed in the supplement.

**Taxi Dispatch Based on Cell Demand Data:** In this scenario, state $x_t \in \mathbb{R}^n$ represents the difference between the number of free taxis and waiting passengers at $n$ city sites, so $x_t > 0$ represents idling taxis while $x_t < 0$ represents queued passengers. Control $u_t \in \mathbb{R}^m$ represents how many taxis are dispatched to serve queued passengers. Exogenous input $s_t \in \mathbb{R}^p$ represents how many new passengers join the queue at time $t$. Of course, the taxi service has a historical forecast of $s_t$, but the cellular operator can use city-wide mobility data to *improve* the forecast. Our goal is to regulate $x_t$ to $L = 0$ to neither have waiting passengers nor idling taxis. In the cost function (Eq. 13), we have $\gamma_e = 1, \gamma_s = 100$ and $\gamma_u = 1$ to heavily penalize customer waiting time for long queues. Our simulations use 4 weeks of stochastic cell demand data from Melbourne, Australia from [34].

**Battery Storage Optimization:** Our final scenario is inspired by a closely-related work to ours [3], who consider how a *single* battery must be charged or discharged based on electricity price forecasts. Since our setting involves a vector timeseries $s$, we consider electrical load forecasts from *multiple* markets. Thus, we used electricity demand data from the same PJM operator as in [3], but from multiple markets in the eastern USA [35]. Specifically, state $x_t \in \mathbb{R}^n$ represents the charge on $n$ batteries and control $u_t \in \mathbb{R}^m$ represents how much to charge the battery to meet demand. Timeseries $s_t \in \mathbb{R}^p$ represents the demand forecast at the locations of the $n$ batteries, where $p = n$. In the cost function (Eq. 13), we desire a battery of total capacity $2L$ to reach a set-point where it is half-full, which, as per [3], allows flexibly switching between favorable markets. Further, we set $\gamma_e = \gamma_s = \gamma_u = 1$.

## 5 Evaluation

The goal of our evaluation is to demonstrate that our co-design algorithm achieves near-optimal control cost, but for much smaller representations $Z$ compared to task-agnostic methods.

**Metrics.** We evaluate the following metrics: 1) We quantify the **control cost** for various bottleneck sizes $Z$, relative to the *optimal cost* when ground-truth input $s$ is shared *without* a network bottleneck. 2) To quantify the benefits of sending a representation of size $Z$ compared to the full forecast $\hat{s}_{t:t+H-1}$ of size $pH$, we define the **compression gain** as $\frac{pH}{Z}$. We also compare the minimum bottleneck $Z$ required to achieve within $5\%$ of the optimal cost for all benchmarks. 3) Since the objective of Prob. 1 also incorporates prediction error, we quantify the **MSE forecasting error** for various $Z$.

**Algorithms and Benchmarks.** We test the above metrics on the following algorithms, which represent various instantiations of Alg. 1 for different $\lambda^{\text{F}}$ as well as today's prevailing method of optimizing for prediction MSE. Our algorithms and benchmarks are: 1) **Fully Task-aware ($\lambda^{\text{F}} = 0$):** We co-design with $\lambda^{\text{F}} = 0$ according to Alg. 1 to assess the full gains of compression. 2) **Weighted**: We instantiate Alg. 1 with $\lambda^{\text{F}} > 0$ to assess the benefits of task-aware compression as well as forecasting errors induced by compression. In practice, $\lambda^{\text{F}}$ is user-specified. For visual clarity, we show results for $\lambda^{\text{F}} = 1$ in Fig. 2 since the trends for other $\lambda^{\text{F}}$ mirror those in Fig. 5. 3) **Task-agnostic (MSE)**: Our benchmark learns a forecast $\hat{s}$ to minimize MSE prediction error, which is directly passed to the controller *without* any co-design.

**Forecaster and Controller Models.** We compared forecast encoder/decoders with long short term memory (LSTM) DNNs [36] and simple feedforward networks. We observed similar performance for all models, which we hypothesize is because co-design needs to represent only a small set of *control-relevant* features. We used standard DNN architectures, hyperparameters, and the Adam optimizer, as further detailed in the supplement. Our code and data are publicly available at `https://github.com/chengjiangnan/cooperative_networked_control`.

### 5.1 Linear Dynamics

We now evaluate our algorithms on the IoT, taxi scheduling, and battery charging scenarios described in Sec. 4. Our results on a *test* dataset are depicted in Fig. 2, where each column corresponds to a real dataset and each row corresponds to an evaluation metric, as discussed below.

**How does compression affect control cost?** The first row of Fig. 2 quantifies the control cost $J^c$ for various compressed representations $Z$. The optimal cost, in a dashed black line, is an unrealizable lower-bound cost when the controller is given the true future $s_{t:t+H-1}$ without any forecast error. The vertical bars show the distribution of costs across several test rollouts, each with different timeseries **s**. Our key result is that our task-aware scheme (orange) achieves within $5\%$ of the optimal cost, but with a small bottleneck size $Z$ of $4$, $4$ and $2$ for the IoT, traffic, and battery datasets, respectively. This corresponds to an absolute compression gain of $15\times$, $15\times$, and $96\times$ for each dataset. In contrast, with the same bottleneck sizes, a competing task-agnostic scheme (blue) incurs at least $25\%$ more control cost than our method.

Moreover, for the IoT and battery datasets, the task-agnostic benchmark requires a large bottleneck of $Z = 35$ and $Z = 11$, leading our approach to transmit $88\%$ and $82\%$ less data respectively. Strikingly, even for a large representation of $Z = 60$, a task-agnostic scheme incurs $100\%$ more cost than the optimal for the cell traffic dataset. This is because the cost function is highly sensitive to shortages with $\gamma_s \gg \gamma_e$, which is not captured by simply optimizing for *mean* error. To clearly see the trend

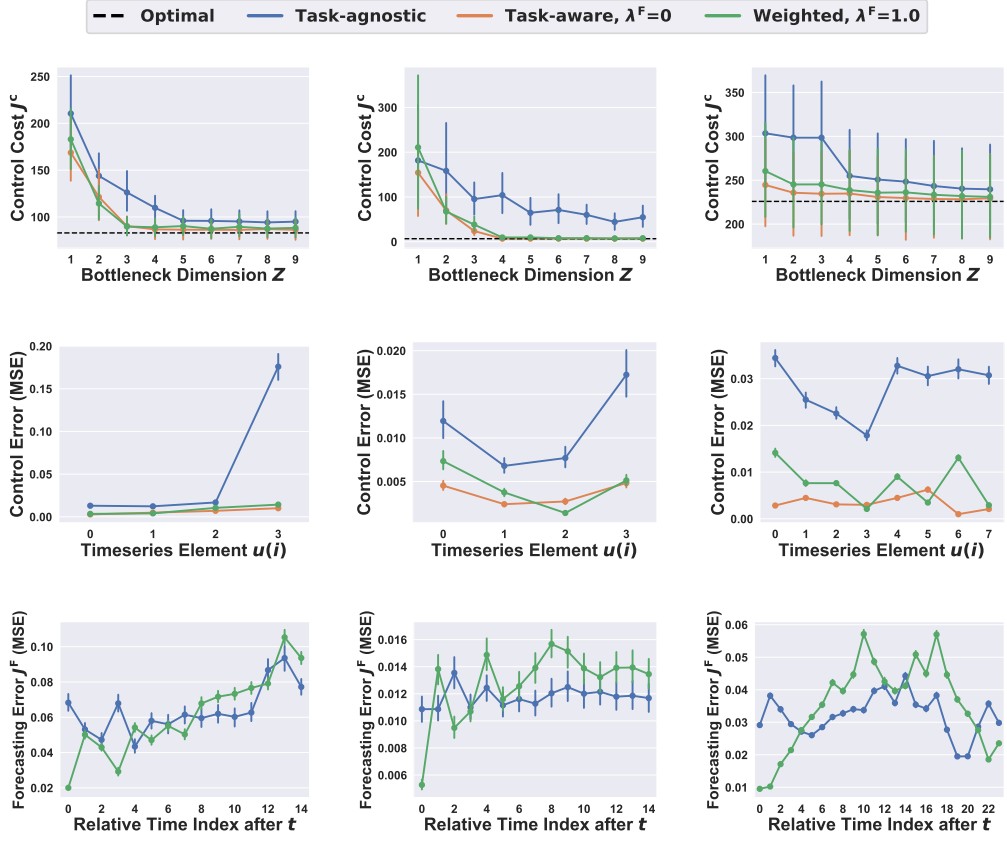

Figure 2: **Real-world dataset results:** From left to right, the columns correspond to smart factory regulation from IoT sensors, taxi dispatching with cell demand, and battery storage optimization. **(Row 1)** Co-design achieves lower cost $J^c$ for smaller bottlenecks $Z$ compared to task-agnostic methods. **(Row 2)** We also achieve lower error for each dimension $i$ of the vector control, $u(i)$, plotted for a highly-compressed $Z = 3$. **(Row 3)** Co-design heavily reduces forecasting errors for initial horizons that are especially important for MPC's decision-making.

in Fig. 2, we only plot until $Z = 9$, but ran the experiments until $Z = 60$. Our weighted approach (green) requires a marginally larger representation than the purely task-aware approach ($\lambda^F = 0$) since it should minimize both control and forecast error.

**Does co-design reduce control errors?** We now investigate how the compression benefits of co-design arise. Given the stochastic nature of all our real world datasets, all prediction models inevitably produce forecasting error, which in turn induce errors in selecting controls. However, the key benefit of co-design methods is they explicitly model and account for how MPC chooses controls based on noisy forecasts $\hat{s}$, and are thus able to minimize the control error, which we now quantify.

As defined in Sec. 3.1, for any state $x_t$, $u_t$ is the optimal first MPC control given perfect knowledge of $s_{t:t+H-1}$, while $\hat{u}_t$ is MPC's actual enacted control given a noisy forecast. Then, the *control errors* across various control dimensions $i$ are the MSE error $||u_t(i) - \hat{u}_t(i)||_2^2$ between optimal control $u_t(i)$ and $\hat{u}_t(i)$. The second row of Fig. 2 clearly shows that our task-aware and weighted methods (orange and green) achieve lower *control* error on all three datasets.

**Why does co-design yield task-relevant forecasts?** To further show that our co-design approach reduces forecasting error for the purposes of an ultimate control task, we show forecasting errors across various time horizons in the third row of Fig. 2. As argued in the previous section, all forecasting models produce prediction error. However, a task-agnostic forecast (blue) roughly equally distributes prediction error across the time horizon $t$ to $t + H - 1$. In stark contrast, the weighted co-design approach (green) drastically reduces prediction errors in the *near future* since MPC enacts

the first control $u_t$ and then re-plans on a rolling horizon. Of course, the full forecast $\hat{s}_{t:t+H-1}$ matters to enact control plan $\hat{u}_{t:t+H-1}$, but the cost is most sensitive to the initial forecast and control errors in our MPC scenarios. For visual clarity, we present forecast errors of the fully task-aware approach ($\lambda^F = 0$) in the supplement, since the errors are much larger than the other two methods.

## 5.2 Nonlinear Dynamics with Transition Noise

To illustrate that our co-design approach works well for systems with **nonlinear and stochastic** dynamics, we also provide a nonlinear example concerning an idealized mobile video streaming scenario. In this application, a mobile video client stores a buffer of video segments and must choose a video quality to download for the next segment of video. The goal is to maximize the quality of video while minimizing video stalls, which occur when the buffer under-flows while waiting for a segment to be downloaded. Here, state $x_t$ represents the buffer of stored video segments, control $u_t$ is segment quality, and $s_t$ is network throughput. The nonlinear dynamics are $x_{t+1} = [x_t - u_t \oslash s_t]_+ + L_x + \eta_t$, where $\oslash$ represents element-wise division, $L_x$ is the increase in stored video for each download, and $\eta_t$ is Gaussian transition noise. The cost aims to keep a positive buffer and have high video

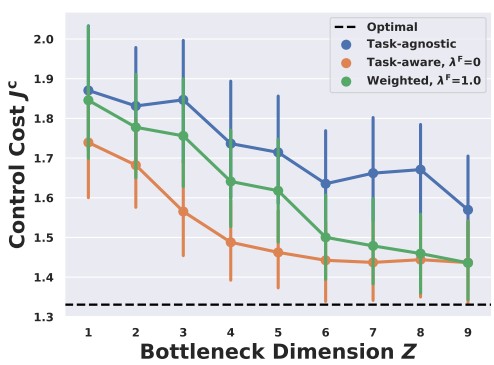

Figure 3: Co-Design Results with *Nonlinear* Dynamics and Transition Noise.

quality: $J^c(\mathbf{x}, \mathbf{u}) = \sum_{t=0}^{T} \gamma_x ||x_t - L_x||_2^2 + \sum_{t=0}^{T-1} \gamma_u ||u_t - L_u||_2^2$.

Fig. 3 clearly shows our approach works quite well for a nonlinear scenario with transition noise, which complements the three linear examples in Sec. 5.1. In the above experiments, the parameters are: $T = 60$, $W = H = 15$, $m = n = p = 4$, $\gamma_x = 0.25$, $\gamma_u = 1$, $L_x = 0.5 \times \mathbb{1}_n$, $L_u = 0.2 \times \mathbb{1}_m$.

**Limitations:** Our work does not automatically learn the optimal bottleneck size $Z$ that minimizes control cost nor necessarily learn a human-interpretable latent representation.

## 6 Conclusion

Society is rapidly moving towards "smart cities" [37, 38], where smart grid and 5G wireless network operators alike can share forecasts to enhance external control applications. This paper presents a preliminary first step towards this goal, by contributing an algorithm to learn task-relevant, compressed representations of timeseries for a control objective. Our future work will center around privacy guarantees that constrain learned representations to filter personal features, such as individual mobility patterns. Further, we want to certify our algorithm does not reveal proprietary control logic or private internal states of the downstream controller. While recent work has addressed how to value datasets for supervised learning [39, 40], a promising extension of our work is to price timeseries datasets for cooperative control in a data-market. Indeed, our ability to gracefully trade-off control cost with data exchange lends itself to an economic analysis.

## 7 Funding Disclosure

This material is based upon work supported by the National Science Foundation under Grant No. 2133481 and 2133403. Any opinions, findings, and conclusions or recommendations expressed in this material are those of the author(s) and do not necessarily reflect the views of the National Science Foundation.

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
