# A    Appendix

## A.1    Time Horizon

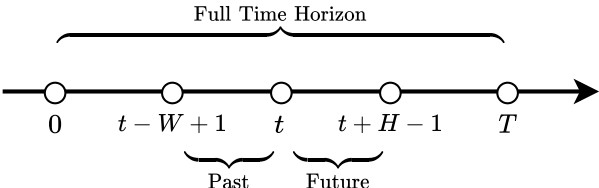

Figure 4: Time horizon illustration.

Fig. 4 illustrates the time horizon of the problem we consider in Sec. 2.

## A.2    Input-driven LQR with Transition Noise

Suppose at each time interval $t$, we add a noise vector $w_t \in \mathbb{R}^n$ with mean zero and covariance $\Sigma_{ww}$ to the dynamics:

$$x_{t+1} = Ax_t + Bu_t + Cs_t + w_t, \tag{14}$$

Then Eq. 4 becomes

$$x_{i+1} = A^{i+1}x_0 + \boldsymbol{M_i}\mathbf{u} + \boldsymbol{N_i}\mathbf{s} + \boldsymbol{P_i}\mathbf{w} \tag{15}$$

where $\mathbf{w} := w_{0:H-1}$ and $\boldsymbol{P_i} = \begin{bmatrix} A^i & A^{i-1} & \cdots & I & \mathbf{0} \end{bmatrix} \in \mathbb{R}^{n \times nH}$.

And hence Eq. 5 becomes

$$J^c(\mathbf{u}; x_0, \mathbf{s}) = \mathbf{u}^\top \underbrace{(\boldsymbol{R} + \sum_{i=0}^{H-1} \boldsymbol{M_i}^\top Q\boldsymbol{M_i})}_{\boldsymbol{K}}\mathbf{u} + 2\underbrace{[\sum_{i=0}^{H-1} \boldsymbol{M_i}^\top Q(A^{i+1}x_0 + \boldsymbol{N_i}\mathbf{s} + \boldsymbol{P_i}\mathbf{w})]^\top}_{\boldsymbol{k}(x_0, \mathbf{s})}\mathbf{u} + \tag{16}$$

$$\underbrace{\sum_{i=0}^{H-1}(A^{i+1}x_0 + \boldsymbol{N_i}\mathbf{s} + \boldsymbol{P_i}\mathbf{w})^\top Q(A^{i+1}x_0 + \boldsymbol{N_i}\mathbf{s} + \boldsymbol{P_i}\mathbf{w})}_{\text{independent of } \mathbf{u}}, \tag{17}$$

that is,

$$\mathbb{E}_\mathbf{w}[J^c(\mathbf{u}; x_0, \mathbf{s})] = \mathbf{u}^\top \underbrace{(\boldsymbol{R} + \sum_{i=0}^{H-1} \boldsymbol{M_i}^\top Q\boldsymbol{M_i})}_{\boldsymbol{K}}\mathbf{u} + 2\underbrace{[\sum_{i=0}^{H-1} \boldsymbol{M_i}^\top Q(A^{i+1}x_0 + \boldsymbol{N_i}\mathbf{s})]^\top}_{\boldsymbol{k}(x_0, \mathbf{s})}\mathbf{u} + \tag{18}$$

$$\underbrace{\sum_{i=0}^{H-1}(A^{i+1}x_0 + \boldsymbol{N_i}\mathbf{s})^\top Q(A^{i+1}x_0 + \boldsymbol{N_i}\mathbf{s}) + \mathbb{E}_\mathbf{w}[(\boldsymbol{P_i}\mathbf{w})^\top Q\boldsymbol{P_i}\mathbf{w}]}_{\text{independent of } \mathbf{u}}. \tag{19}$$

Notice that $\mathbf{w}$ only affects the constant term, which is independent of $\mathbf{u}$. Therefore, the analysis after Eq. 5 still holds.

## A.3    Additional Explanations on the Proof of Proposition 1

Here, we provide some additional explanations on the proof of Proposition 1, which are not included in the main paper due to space limits.

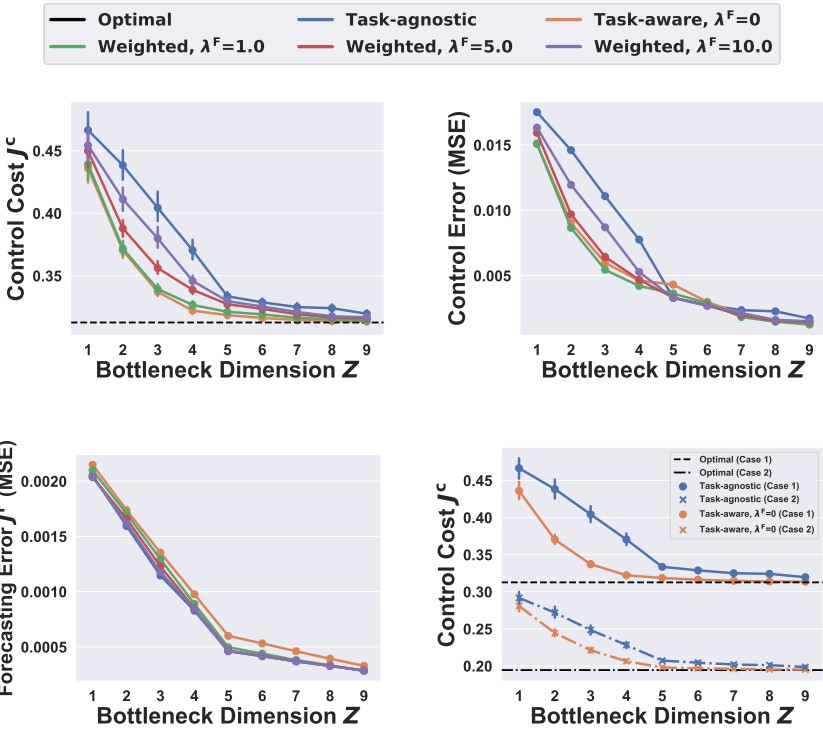

Figure 5: **Analytic results for linear control:** (a) By only representing information salient to a control task, our co-design method (orange) achieves the optimal control cost with $43\%$ less data than a standard MSE approach ("task-agnostic", blue). Formal definitions of all benchmarks are in Sec. 5. (b-c) By weighting prediction error by $\lambda^{\mathrm{F}} > 0$, we learn representations that are compressible, have good predictive power, and lead to near-optimal control cost (*e.g.* $\lambda^{\mathrm{F}} = 1.0$). (d) For the *same* timeseries $\mathbf{s}$, two different control tasks require various amounts of data shared, motivating our task-centric representations.

1) **Positive definite matrix.** $\Psi + \lambda^{\mathrm{F}} I$ ($\lambda^{\mathrm{F}} > 0$) is positive definite because, $(\Psi + \lambda^{\mathrm{F}} I)^{\top} = \Psi + \lambda^{\mathrm{F}} I$, and $J^{\mathrm{tot.}} \geq \lambda^{\mathrm{F}} ||\hat{\mathbf{s}} - \mathbf{s}||_2^2 > 0$ for any $\hat{\mathbf{s}} \neq \mathbf{s}$.

2) **Eigen-decomposition.** The eigen-decomposition of $\Psi + \lambda^{\mathrm{F}} I$ is $Y \Lambda Y^{-1}$, where $Y \in \mathbb{R}^{pH \times pH}$ and the columns of $Y$ are the normalized eigen-vectors of $\Psi + \lambda^{\mathrm{F}} I$, and $\Lambda \in \mathbb{R}^{pH \times pH}$ is the diagonal matrix whose diagonal elements are the eigenvalues of $\Psi + \lambda^{\mathrm{F}} I$. Since $\Psi + \lambda^{\mathrm{F}} I$ is symmetric, $Y$ is also orthogonal, i.e., $Y^{-1} = Y^{\top}$. So $Y \Lambda Y^{-1} = Y \Lambda Y^{\top}$.

3) **Inverse matrix.** The matrix $\Lambda^{\frac{1}{2}} Y^{\top}$ is invertible because $\Psi + \lambda^{\mathrm{F}} I$ is positive definite and its eigenvalues are all positive.

## A.4 Details on the LQR Simulations

Here we provide further details on the two LQR simulations mentioned in Sec. 3.1. In both of the simulations, vector timeseries $s$ has log, negative exponential, sine, square, and saw-tooth functions superimposed with a Gaussian random walk noise process.

### A.4.1 Basic LQR Simulation (Fig. 5)

1) **Dynamics:**

$$x_{t+1} = x_t + u_t - C s_t$$

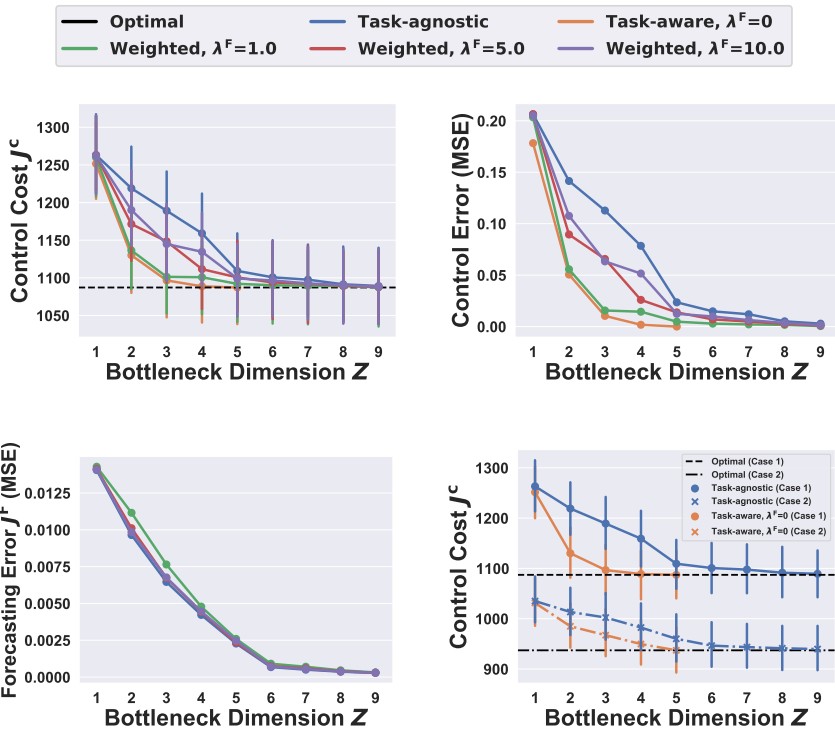

Figure 6: **Linear control with MPC:** We repeat our analysis of input-driven LQR, but solve the problem in a receding horizon manner with forecasts for $H < T$ as discussed in Section 3.1 and Figure 5. (a) By only representing information salient to a control task, our co-design method (orange) achieves the optimal control cost with $60\%$ less data than a standard MSE approach ("task-agnostic", blue). Formal definitions of all benchmarks are in Sec. 5. (b-c) By weighting prediction error by $\lambda^F > 0$, we learn representations that are compressible, have good predictive power, and lead to near-optimal control cost (*e.g.* $\lambda^F = 1.0$). The forecasting error of the task-aware scheme (orange) is much larger than the rest and thus not shown in the zoomed-in view. (d) For the *same* timeseries **s**, two different control tasks require various amounts of data shared, motivating our task-centric representations.

2) **Cost function:**

$$J^c = \frac{1}{1000}\left(\sum_{t=0}^{H}||x_t||_2^2 + \sum_{t=0}^{H-1}||u_t||_2^2\right)$$

3) **Parameters:** $H = 20$; $n = m = p = 5$; $C = \mathrm{diag}(1, 2, \cdots, 5)$ for Fig. 5(a)-5(c) and Fig. 5(d) top, and $C = \mathrm{diag}(1.5, 2, \cdots, 3.5)$ for Fig. 5(d) bottom.

As per Proposition 1, we solve a simple low-rank approximation problem per bottleneck $Z$ to obtain the optimal encoder $E$, decoder $D$, and use Eqs. 9-10 to obtain the control and prediction costs. Clearly, our co-design algorithm (orange) outperforms a task-agnostic approach (blue) that simply optimizes for MSE.

### A.4.2 LQR Simulation with MPC (Fig. 6)

1) **Dynamics:**

$$x_{t+1} = x_t + u_t - Cs_t$$

2) **Cost function:**

$$J^c = \sum_{t=0}^{T}||x_t||_2^2 + \sum_{t=0}^{T-1}||u_t||_2^2$$

3) **Parameters:** $T = 100$, $W = H = 15$; $n = m = p = 5$; $C = \mathrm{diag}(1, 2, \cdots, 5)$ for supplement Fig. 6(a)-6(c) and Fig. 6(d) top, and $C = \mathrm{diag}(3, 3, \cdots, 3)$ for Fig. 6(d) bottom.

## A.5  IoT Data Collection

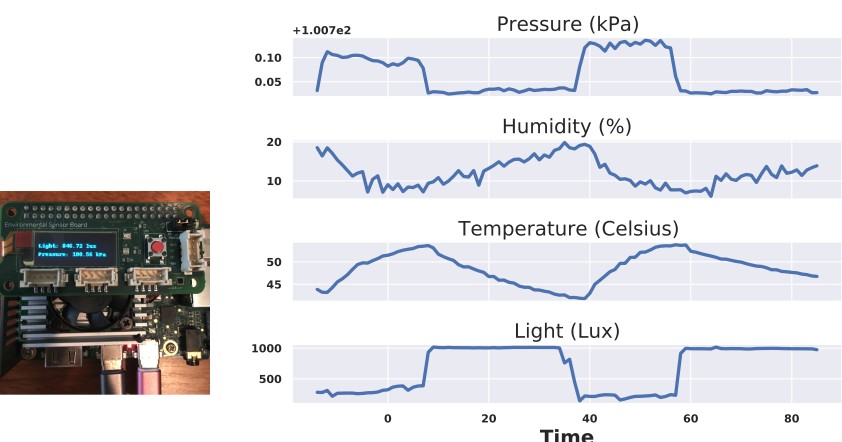

Figure 7: Environmental sensor on the Google Edge TPU (left) and example stochastic timeseries (right).

Fig. 7 shows the environmental sensor board (connected to an Edge TPU DNN accelerator) and an example of collected stochastic timeseries for our IoT data.

## A.6  Detailed Evaluation Settings

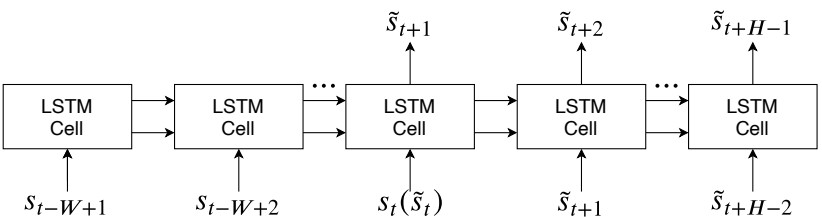

Figure 8: LSTM timeseries network

We now provide further details on Sec. 5 by summarizing the settings of our evaluation.

### A.6.1  Forecaster, Controller & Data Scaling

**Basic Forecaster Settings.** In all three scenarios, the encoder parameters $\theta_e$ are responsible for *both* forecasting and compression. We first have a forecasting model that first provides a full-dimensional forecast $\tilde{s}_{t:t+H-1}$, and then adopts simple linear encoder $E \in \mathbb{R}^{Z \times pH}$ to yield $\phi_t$. The combination of the forecasting model's parameters and encoder $E$ constitute $\theta_e$. Then, a linear decoder $D \in \mathbb{R}^{pH \times Z}$ eventually produces decoded forecast $\hat{s}_{t:t+H-1}$. The model used to provide full-dimensional forecast $\tilde{s}_{t:t+H-1}$ varies case by case, as described subsequently.

**Smart Factory Regulation with IoT Sensors.** For forecasting, we adopt an LSTM timeseries network, as shown in Fig. 8, with $W + H - 2$ cells and hidden size 64. The parameters associated with the forecaster and controller are set as follows: $T = 72$, $W = H = 15$, $n = m = p = 4$;

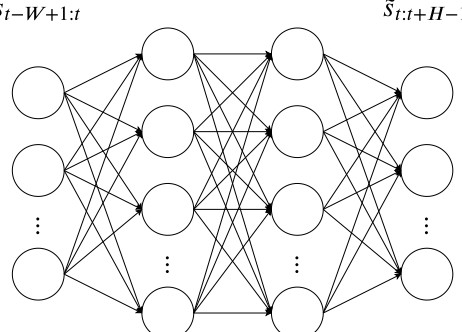

$$s_{t-W+1:t} \qquad\qquad\qquad \tilde{s}_{t:t+H-1}$$

Figure 9: 2-hidden-layer feedforward network

$u_{\min} = -0.95 \times \mathbb{1}_p, u_{\max} = 0.95 \times \mathbb{1}_p, \gamma_e = \gamma_s = \gamma_u = 1$. Further, we scale $s_t(i)$ to be within $[-1, 1], \forall i$.

**Taxi Dispatch Based on Cell Demand Data.** For forecasting, we adopt a 2-hidden-layer feedforward network, as shown in Fig. 9, with hidden size $64$ and ReLu activation. The parameters associated with the forecaster and controller are set as follows: $T = 32$, $W = H = 15$, $n = m = p = 4$; no constraint on $u_t$, and $\gamma_e = 1, \gamma_s = 100, \gamma_u = 1$. Further, we scale $s_t(i)$ to be within $[0, 1], \forall i$.

**Battery Storage Optimization.** For forecasting, we adopt a 2-hidden-layer feedforward network, as shown in Fig. 9, with hidden size $64$ and ReLu activation. The parameters associated with the forecaster and controller are set as follows: $T = 122$, $W = H = 24$, $n = m = p = 8$; no constraint on $u_t$, and $\gamma_e = \gamma_s = \gamma_u = 1$. Further, we scale $s_t(i)$ to be within $[0, 1], \forall i$.

We observed similar performance for feedforward networks and LSTMs since the crux of our problem is to find a small set of task-relevant features for control.

### A.6.2 Training

Table 1: Train/Test Timeseries, Training Epochs and Runtime.

| DATASET | TRAIN/TEST TIMESERIES | TRAINING EPOCHS | RUNTIME |
|---|---|---|---|
| IoT | 30/30 | 1000 | < 96 HRS |
| CELL | 17/17 | 1000 | < 48 HRS |
| BATTERY | 15/15 | 2000 | < 1 HR |

Our evaluation runs on a Linux machine with 4 NVIDIA GPUs installed (3 Geforce and 1 Titan). Our code is based on Pytorch. We use the Adam optimizer and learning rate $10^{-3}$ for all the evaluations. The number of train/test timeseries[3], training epochs, and resulting runtime are summarized in Table 1. The IoT dataset is provided in our code release and it does not have any personally identifiable or private information. The publicly-available electricity and cellular datasets did not have a stated license online.

### A.7 Further Analysis on the Evaluation Results

For better understanding of the differences between different schemes, we give further analysis on our evaluation results in Sec. 5.

**Why does co-design yield task-relevant forecasts? (Continued)**

We further contrast the prediction errors made by task-agnostic and co-design approaches in the heatmaps of Fig. 10. In each heatmap, the x-axis represents the future time horizon, while the y-axis

---

[3]With MPC, each timeseries corresponds to $T$ samples, such as $T = 72$ for the IoT scenario.

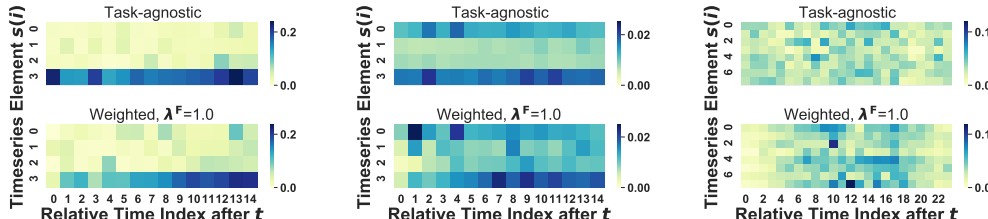

Figure 10: **Forecasting error comparison: task-agnostic vs. weighted scheme.** From left to right, the columns correspond to smart factory regulation from IoT sensors, taxi dispatching with cell demand, and battery storage optimization. The heatmaps show how co-design minimizes errors on timeseries elements $s(i)$ and forecast horizons that are salient for the control task when $Z = 3$.

represents forecasting errors across various dimensions of timeseries $s$, denoted by $s(i)$. Clearly, a weighted approach significantly reduces prediction error for near time-horizons, which is most pronounced for the battery dataset.

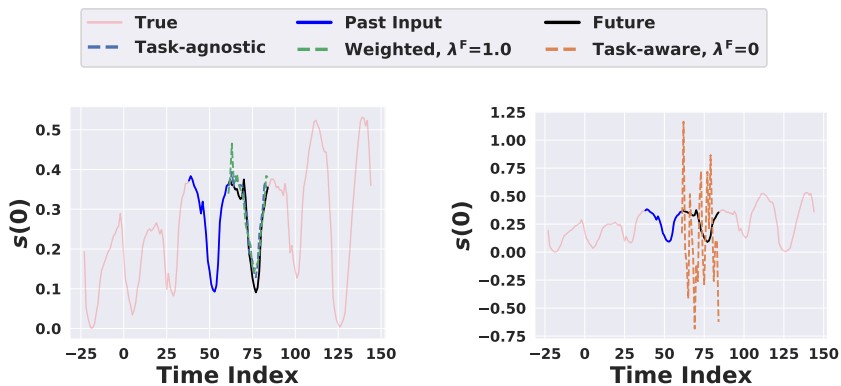

Figure 11: **Forecast comparison: task-agnostic/weighted schemes vs. task-aware scheme.** Example forecasts for the battery charging scenario at $t = 84$ when $Z = 9$, for both our task-agnostic/weighted schemes (left) and the task-aware scheme (right). Clearly, a fully task-aware approach with $\lambda^F = 0$ yields poor predictions since it does not regularize for prediction errors. This motivates our weighted co-design approach on the left.

**The fully task-aware ($\lambda^F = 0$) scheme is good for control but poor for forecasting.**

Fig. 11 compares the time-domain forecasts given by task-agnostic/weighted scheme and task-aware scheme. Note that the timeseries starts at $t = -W + 1 < 0$ because $s_{-W+1:0}$ is needed at $t = 0$. While the task-agnostic and weighted scheme make reasonable forecasts, the task-aware scheme focuses solely on improving the task-relevant control and imposes no penalties on the forecasting error, leading to poor forecasts. This motivates our weighted approach which balances the control cost and forecasting error.

**Small $Z$ (e.g., $Z = 4$) produces coarse forecasts, which are suitable for good control performance.**

Fig. 12, Fig. 13 and Fig. 14 present the time-domain forecasts with different bottleneck dimensions $Z$ for IoT, taxi scheduling, and battery charging scenarios, respectively. In general, for small $Z$ (e.g., $Z = 4$), the task-agnostic scheme makes noisy forecasts which provides room for our weighted

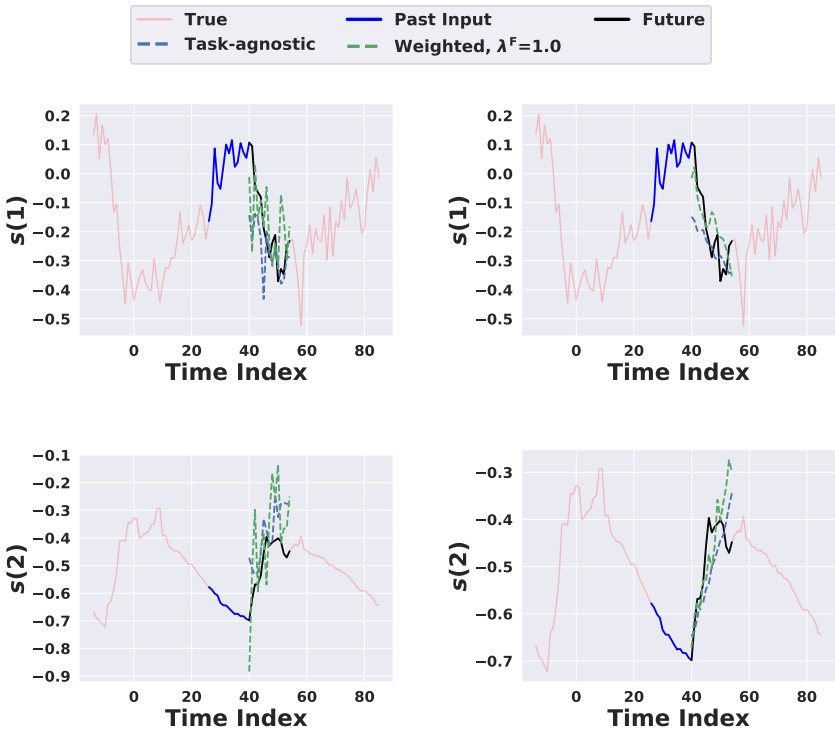

Figure 12: **Example forecasts (IoT sensors).** Example forecasts at $t = 40$ when $Z = 4$ (left) and $Z = 9$ (right). Clearly, the predictions are more accurate and smooth when $Z = 9$. However, with a smaller bottleneck of $Z = 4$ (left), we achieve near-optimal control performance since we capture task-relevant features with a coarse forecast that captures high-level, but salient, trends.

scheme to improve the control cost by considering a task-relevant objective. For large $Z$ (e.g., $Z = 9$) both the task-agnostic and weighted scheme make smooth forecasts[4].

**The state evolution of our task-aware/weighted scheme is closer to the optimal trace.**

Fig. 15 shows the example state evolution of $x(2)$ for the three scenarios. Importantly, the black trace corresponds to an *unrealizable* baseline with the lowest cost since it assumes *perfect* knowledge of $\mathbf{s}$ for the future $H$ steps. We can see that our task-aware and weighted scheme have state evolution traces closer to the optimal trace than the competing task-agnostic scheme. This further explains why task-aware and weighted schemes can yield a near-optimal cost for small $Z$ while the task-agnostic benchmark cannot.

---

[4]The trend is less prominent for the taxi scheduling scenario, because the cell demand itself is rapidly-changing and highly-stochastic.

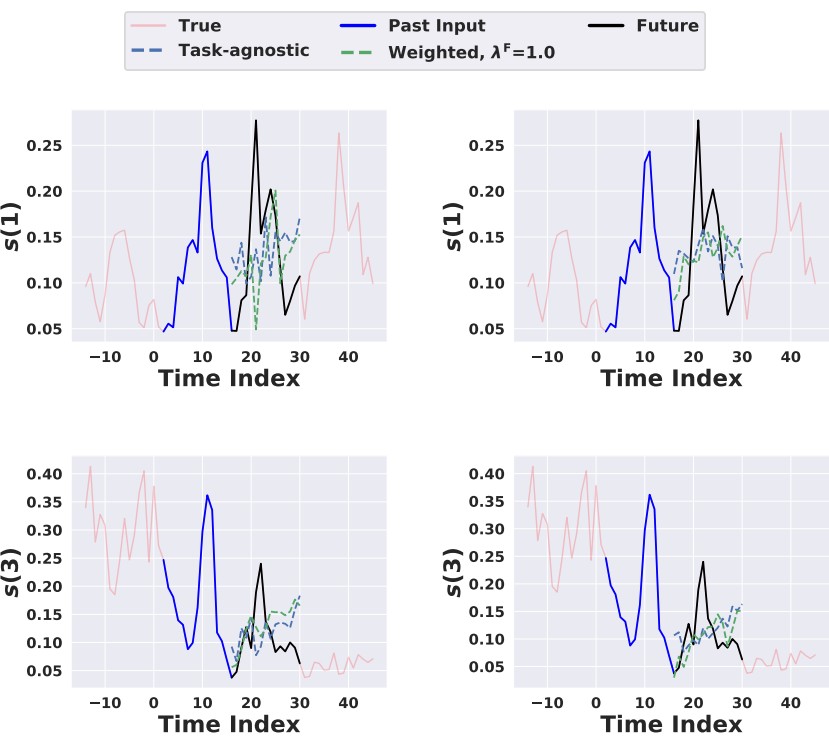

Figure 13: **Example forecasts (taxi scheduling).** Example forecasts at of at $t = 16$ when $Z = 4$ (left) and $Z = 9$ (right). This scenario had the worst prediction errors since the cell data is highly stochastic.

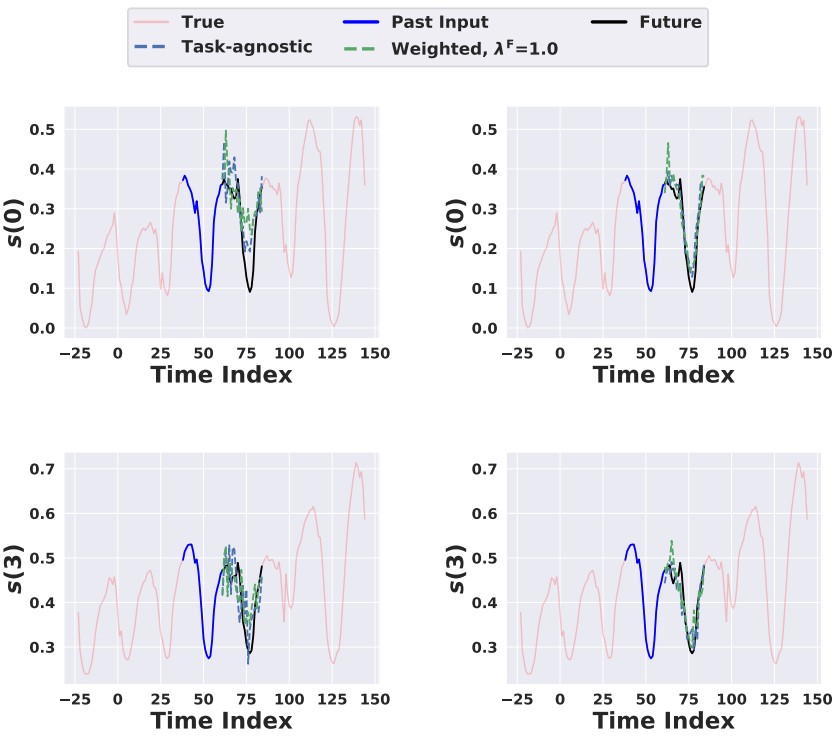

Figure 14: **Example forecasts (battery charging).** Example forecasts at $t = 84$ when $Z = 4$ (left) and $Z = 9$ (right). As before, the predictions are more accurate and smooth when $Z = 9$. However, with a smaller bottleneck of $Z = 4$ (left), we achieve near-optimal control performance since we capture task-relevant features with a coarse forecast that captures high-level, but salient, trends.

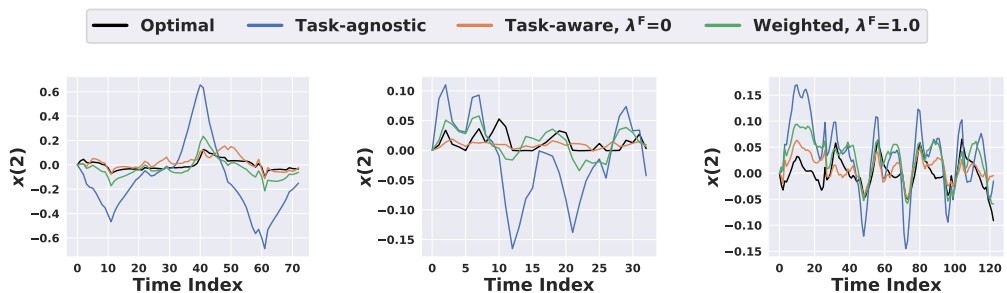

Figure 15: Example evolution of $x(2)$ when $Z = 4$, for IoT (top), taxi scheduling (middle) and battery charging (bottom) scenarios, respectively. Clearly, our co-design approach has state evolutions closer to the unrealizable optimal solution (black) which assumes *perfect* forecasts.