# OpenReview forum: "Data Sharing and Compression for Cooperative Networked Control"
_NeurIPS.cc/2021/Conference — NeurIPS 2021 Poster_

### Official Review · Reviewer_ZtkH · 2021-07-02

**Rating:** 6
**Confidence:** 4

**Summary:**

The paper discusses an approach for deterministic optimal control under estimates of covariate states.

**Main Review:**

Overall, I found the paper a good read and easy to follow. However, I am not sure if there is a significant merit to the paper, as the theoretical work seems to me well known and pretty straightforward. I would have either hoped for a bigger focus in the experimental section, which I would have expected to include a real world control scenario or for more work in a theoretical direction for example by including stochastic dynamics.

Originality: The task overall is not known to me and I am not sure if there are a lot of real world applications for the scenario. The methods presented in the work are pretty standard and there are not many new theoretical insights presented. To me it was not completely clear how the work positions itself in the field w.r.t. related work.

Quality: I did not find any technical mistakes and the mathematics are correct as far as I could check.

Clarity: The submission is nicely written, overall easy to follow and well organized.

Significance: I am not completely sure if the results are very significant.

Aspects I found which were missing from a theoretical standpoint are:
- The work only discusses a deterministic system and therefore no noisy dynamics. Additionally, a probabilistic model description especially for the estimation of the forecast state is probably very favorable in many scenarios.
-  For the forecast only point estimates are considered. I think a Bayesian estimate would be more appropriate. As this work does not include any uncertainty about the estimates. However, a controller should probably balance its control decisions if estimates are uncertain.
- Something I did not completely understand was why the controller for $\hat{u}$ should have the same parametric form as the one for $u$. What is the motivation for this design, similar to certainty equivalence?

Things I missed from the experimental section:
- Why is the work only evaluated for the very simple dynamical system? For a synthetic evaluation I would have expected also non-linear system dynamics.
- I would have appreciated if the system would have been evaluated for a real world control problem.
- For the  closed form solutions in the linear case, I would have appreciated an evaluation.


**Time Spent Reviewing:**

6

---

> ### Author Response · Authors · 2021-08-10
> **Response to Reviewer ZtkH**
>
> We thank the reviewer for their thoughtful, constructive comments. Actually, two of the reviewer’s key concerns are already addressed experimentally in our paper:
>
> 1. Non-linear, stochastic dynamics (Main paper lines 264-266): We show strong results in Appendix A.1 for the nonlinear setting with stochastic dynamics, for an application of mobile video streaming. The only reason we placed this in the appendix was due to space limits and we preferred all three scenarios in the main paper to be of societal importance in smart cities (taxi routing, smart batteries, IoT sensors in factories).
> 2. Evaluation of Closed Form Solutions for Linear Case (Main paper lines 204 and 211): Appendix Figure 5 shows exactly what the reviewer is asking for. Indeed the plot title is ‘Analytical Results for linear control’ and they mirror the strong gains we found in our real-world datasets.
>
> Further, the reviewer states that the ‘theoretical work is well-known’. Actually, we exhaustively searched relevant papers on tele-operation, networked control etc. and our results are novel. The key novelty is that we use a weighted control and reconstruction loss, which leads to a different task-driven encoder/decoder through the SVD. This is in stark contrast to today’s approaches from information theory, rate-distortion theory, and compressed sensing that largely optimize for pure reconstruction error.
>
> As for significance, mechanisms for data compression and data sharing for control are of prime importance in future city-wide 5G networks, smart factories with IoT sensors, and power grids.
> These are the domains that inspire our experimental results.
>
> The reviewer is correct that incorporating Bayesian forecasts would be very theoretically interesting. We left this out since we talked to network operators before starting this research, who cannot run Bayesian forecasts due to the computational complexity of generating calibrated uncertainty estimates for terabytes of streaming data. However, we appreciate the reviewer’s insightful ideas for future theoretical analysis and will add it to the paper’s discussion.
>
> Finally, the controller for $\hat{u}$ has the same parametric form as $u$ due to inspiration from certainty-equivalence. The reviewer is correct that, in principle, we could use Bayesian forecasts and look at risk-sensitive control. However, as stated above, the computational complexity of doing this on terabytes of data could potentially be too high for real network operators with compute constraints.

---

> > ### Author Response · Authors · 2021-08-28
> > **Key requested experiments are in the appendix - happy to clarify further.**
> >
> > Thanks for the review. We wanted to check if you had any questions on our experiments in the appendix (for nonlinear dynamics and evaluation of the closed form results for LQR). These experiments in A.1 and Fig A5 cover some of the experimental questions you had in the original review, and we are happy to clarify further.

---

> > > ### Comment · Reviewer_ZtkH · 2021-08-30
> > > **Thank you for your clarifications.**
> > >
> > > Thank you for your clarifications.
> > > I acknowledge the fact that some of my concerns are addressed in the appendix and I increased my score to a 6 and I will not fight for a rejection.
> > > I appreciate the extended evaluation for the linear case and the evaluation for the non-linear dynamics.
> > > In my opinion the method is not really tackling the stochastic case, as the cost function is not in terms of an average cost criterion.
> > > In general, I think that a thorough discussion of a probabilistic description is missing in the paper .
> > > Overall I still stand by my review and am of the opinion that the contribution is a bit marginal.

---

> > > > ### Author Response · Authors · 2021-08-30
> > > > **Thanks for looking at the appendix**
> > > >
> > > > Thank you for increasing our score and the valuable comments. We will definitely add a discussion of the stochastic case in the paper.

---

### Official Review · Reviewer_vpGi · 2021-07-16

**Rating:** 7
**Confidence:** 3

**Summary:**

This paper presents a solution to learn succinct, compressed time-series embeddings, which also considers downstream modular controller’s task objective.

Contributions:
1. novel problem: learning compressed time series representations which are also tailored to the downstream control tasks.
2. Analytic compression results on LQR control for illustrations on the problem.
3. Experiments on real-world datasets

**Limitations And Societal Impact:**

The authors pointed out that they do not automatically learn the optimal bottleneck size that minimizes control cost nor necessarily learns a human-interpretable latent representation.

**Main Review:**

Originality: This paper studies a novel task that learns time series representations for the downstream control tasks. This paper discussed clearly how this work differs from previous contributions on page 3,  with the most relevant paper cited.

Quality: The submission is technically sound and the claims are well supported by theoretical analysis on the  LQR problem. Then the paper expands to more general settings like DNN forecasters.

Clarity: The submission is clearly written and well organized.

Significance: The results are important, not only from the perspective of improvements on baselines but also from the perspective of possible application domains. This paper discussed the diverse application scenarios like smart factory on IoT sensors, taxi dispatch, and battery storage.

**Time Spent Reviewing:**

2

---

> ### Author Response · Authors · 2021-08-10
> **Response to Reviewer vpGi**
>
> We thank the reviewer for appreciating our paper and the implications for application domains in future smart cities. If this paper is published, we hope it will spur collaborations with 5G network operators to obtain more city-scale data to stress-test our methods. We are happy to answer any questions during the rebuttal period.

---

### Official Review · Reviewer_zhDr · 2021-07-17

**Rating:** 6
**Confidence:** 3

**Summary:**

This paper studies sending compressed representations for cooperative control by creating task-driven representations that are co-designed with the control task. They present theoretical compression results (of LQR being low-rank compression in prop 1) and go on to study non-linear cellular, IoT, and electricity load data settings. The idea, summarized in Algorithm 1, is to parameterize encoders/decoders, use the decoded signal for optimization, and optimize the loss in Eq 8 that includes the control loss and a prediction component.

**Limitations And Societal Impact:**

The paper does not significantly discuss these and it could be insightful to discuss scalability limitations on how scaling the bottleneck dimension and control problem sizes impact the overall tractability/learnability.

**Main Review:**

This is a well-motivated and timely study of end-to-end learning for cooperative control. They take the existing pieces of differentiable control and end-to-end model learning and theoretically and empirically study them in a new cooperative context. To the best of my knowledge, the theoretical results in the LQR setting are novel, and the experiments in Fig 2 validate the idea that including the downstream signal enables more efficient latent representations of the data to be learned. The battery storage optimization task from [3] is reasonable extended from the single to multiple battery setting. My largest experimental concerns is that the bottleneck dimensions used in Fig 2 are relatively small and only consider dimensions <10. Realistically, the bandwidth in these settings should allow for higher-dimensional data to be transmitted.

**Time Spent Reviewing:**

1 hour

---

> ### Author Response · Authors · 2021-08-10
> **Response to Reviewer zhDr**
>
> We thank the reviewer for appreciating our paper. We believe our method will scale well to large bottleneck sizes and complex datasets. This is because our key insight is to exploit the known dynamics of a model-based controller and compute its sensitivity to forecasting errors, which guides efficient supervised learning of a task-specific encoder/decoder. Our supervised learning methods will scale much more to large dimension bottlenecks compared to learning-based control methods, such as deep reinforcement learning (RL).
>
> Also, we only used small bottleneck dimensions since they were sufficient for our datasets, such as the cell data with ~10 cells in downtown Melbourne. In reality, large cities have over 3000 cell sites and will measure terabytes of data per day. In those settings, we believe our efficient co-design method will make data sharing scalable for important smart city applications. If this paper is published, we hope it will spur collaborations with 5G network operators to obtain more data to stress-test our methods. We are happy to revise the discussion of limitations in our paper.

---

### Official Review · Reviewer_rL7L · 2021-07-18

**Rating:** 6
**Confidence:** 4

**Summary:**

This paper proposes an architecture and gradient-based training for an encoder/decoder scheme to reduce communication of an exogenous signal to a remote optimization-based controller. Such scenarios are given, for example, when a remote location has forecast information (e.g., traffic) that is relevant for an optimization-based controller (e.g., optimal scheduling of taxis). The goal of the paper is to reduce communication of the exogenous, while maintaining reasonable performance. The main idea is to take the control objective into account when training the compression scheme, so that the encoder/decoder is optimized for the task at hand. The claimed contributions are (i) introduction of a novel problem, (ii) analytical results for the LQR case, and (iii) improved performance and compression on real-world data sets.

**Limitations And Societal Impact:**

Yes, IMO.

**Main Review:**

## Originality

1) The components of the presented approach (encoder/decoder for reduced communication, gradient-based end-to-end training, remote control architecture) are not new. Also the idea to take a downstream task into account when designing reduced communication schemes is not entirely new (see next comment). Nonetheless, this reviewer is not aware of the proposed combination of techniques. Thus, I see the main contribution in that combination and the demonstration that this leads to reduced communication.

2) Related work: The discussion of related work is mostly adequate.  However, I was missing a *specific* discussion on the technical results for the LQR case with exogeneous input in Sec. 3.1.   The LQR problem with exogenous inputs has been studied before - so, how are the technical results derived in Sec. 3.1 related to results known in control?  Just to be clear - the combination of results is likely new, but still, it would be good to see a discussion on how the obtained equations, e.g. for the optimal controller, compare to existing works.

I have further suggestions for the discussion of related work:

2.1) The idea to reduce amount of network traffic by transmitting only the data that is relevant for a certain task has also been investigated under the terms of "event-triggered communication/estimation/control" (see surveys Heemels et al, An introduction to event-triggered and self-triggered control; or Miskowicz, Event-based control and signal processing) and, more recently, "event-triggered learning".  For example, Schlueter et al. (Event-triggered Learning for Linear Quadratic Control) also use LQR control cost to trigger learning of a better model when needed. These ideas have also been applied to reduce network communication in wireless sensor systems with savings similar to those reported herein (e.g., Beuchert et al, Overcoming bandwidth limitations in wireless sensor networks by exploitation of cyclic signal patterns: An event-triggered learning approach) . While the mentioned event-triggered approaches emphasize sparsity in time, the manuscript at hand has a different perspective on reducing communication.  Thus, the works are not conflicting (maybe orthogonal?), yet are closely related on the level of motivation and application to reducing communication in control networks. In fact, a combination of the approaches might be interesting for further research.
2.2) When reading the "Related Work" subsection at the beginning of page 2, I was surprised to see no discussion on teleoperation, e.g. in robotics.  The authors later add such a discussion (to some extent), so it might be worth pointing out already at the end of the "related work" subsection, that more discussion will follow after introducing the technical problem in detail.



## Quality

The paper seems technically sound.  Limitations are briefly discussed at the end of Sec. 5., and in the conclusion.  The numerical examples serve to illustrate main claims of the paper.  IMHO, the experimental results could, however, be improved by reporting on nonlinear examples:

3) The numerical examples presented in the main paper all include linear dynamics.  Obviously, these are less challenging then nonlinear problems.   For a nonlinear problem, the reader is deferred to the supplementary material.  I would find it much more convincing if the nonlinear example was part of the main paper.



## Clarity

I found the submission clear.  I only have a minor comment on the abstract:

4) In the abstract, "forecasts are designed without knowledge of a downstream controller’s task objective, and thus simply optimize for *mean* prediction error."  -- I don't see this line of argumentation.  Why does the ignorance of a downstream task give rise to optimizing the MEAN only?  In fact, as one does not know what downstream task will be performed, one might want to optimize for predicting full distributions (even though this is hard, of course).



## Significance

5) As stated above, I see the main contribution in the combination of existing techniques.  To my knowledge, the concrete setting and architecture is novel.  Though, I did not find it too surprising that one can train an encoder/decoder in such a way to save communication. The empirical results demonstrate improvement over more standard ways to train encoder/decoder (without knowledge of the downstream task), yet they don't compare to alternative ways to save communication (e.g., see works mentioned under 2)), so it is hard to judge how significant the communication savings are.

**Time Spent Reviewing:**

~3h

---

> ### Author Response · Authors · 2021-08-10
> **Response to Reviewer rL7L**
>
> We thank the reviewer for their insightful comments, which we now discuss in turn.
>
> Related work - We agree that we can include the tele-operation references earlier in the paper, which we will do in the final version. Our problem setting and LQR results are novel, as discussed after the problem statement. In particular, classical settings of LQR with exogenous disturbances treat the disturbance as Gaussian noise, but we make the insight that they are spatio-temporal patterns in our setting and thus can be efficiently compressed based on their relevance to the ultimate control task.
>
> We thank the reviewer for the references to event-triggered control, which we will add. However, we are aware of event-driven control and did not originally include it because, as the reviewer states, it addresses temporal sparsity while our MPC controllers consistently require a forecast of demand. Thus event-triggered control is quite different from our approach. Our key observation is that streaming, sustained city-wide demand patterns can be compressed based on how relevant spatiotemporal trends are for the ultimate control task.
>
> Nonlinear dynamics example - We can easily add the nonlinear example to the main paper. We chose the examples in the main paper to reflect important applications in smart cities - taxi routing, battery scheduling, and smart factories for a unified exposition.
>
> Abstract - We agree with the reviewer on revising the abstract sentence. We meant that task-agnostic forecasts often minimize mean squared error (MSE), but also could try to capture the full distribution.

---

> > ### Comment · Reviewer_rL7L · 2021-08-26
> > **Answer to authors' response**
> >
> > Thank you for responding to some of my comments.
> >
> > I'm afraid I disagree with the statement "classical settings of LQR with exogenous disturbances treat the disturbance as Gaussian noise". If the additional input is indeed Gaussian noise, this is more commonly termed "process noise".  While indeed, this is a common setting (the linear quadratic Gaussian setting), an "exogenous input" typically denotes a different type of signal, for example, a deterministic disturbance or some other input (might be noise, but does not have to be).  I would be surprised if this setting was never treated in the literature.  For example, a QUICK google search revealed this paper:
> > Singh, Pal "An extended linear quadratic regulator for LTI systems with exogenous inputs", Automatica, 2017
> >
> > Overall, I have to say that I'm not convinced by the response on my criticism and request to see a "*specific* discussion on the technical results for the LQR case with exogeneous input in Sec. 3.1".  It is still not clear to me where the (technical) novelty is in this part.

---

> > > ### Author Response · Authors · 2021-08-27
> > > **2nd Response to Reviewer rL7L**
> > >
> > > Thank you very much for your feedback, which we are happy to discuss. We will first describe the novelty with respect to Singh, Pal Automatica 2017 and then describe the technical novelty.
> > >
> > > We have tried our best to extensively search related literature. We were aware of Singh, Pal 2017 and thought it was not quite related after studying the paper. While (Singh, Pal 2017) and related works discuss input-driven LQR, their setting of an exogenous disturbance is altogether different from ours. First, their input $s_t$ is either stationary with known statistics (e.g., Eq. (65) in their Section 5.2) or purely unpredictable (Section 5.3 of their paper). Second, their input $s_t$ is not subject to a network bottleneck and encoder/decoder, which is the crux of our Prob. 1. Most importantly, their cost automatically penalizes $s_t$ quadratically (Eq. (10), Singh), which is inappropriate in our setting since a controller should only be penalized for erroneous controls due to mis-estimation of $s_t$, not the exogenous $s_t$ itself, which is independent of the controls. In other words, their formulation would directly penalize high cell demand $s_t$ instead of mis-applied controls, which is clearly inappropriate since demand is driven by non-stationary commute patterns. Specifically, our forecasting penalty in Eq 1 ($J^F$) and line 113 penalizes the error in $\hat{s}_t$, not $s_t$ itself. Having said all these, if the reviewer prefers, we can cite Singh, Pal 2017 and explain why it deals with a quite different problem.
> > >
> > > More broadly, the technical novelty of the linear setting arises due to three reasons (mentioned in lines 126-145).
> > >
> > > 1. Non-Stationary $s_t$:  The operator sees a non-stationary, stochastic timeseries $s_t$ (e.g. cell demand), without an analytical process model for standard Kalman Filtering. Specifically, Section 3.1 and Eqs 2 - 10 make no assumption on $s_t$ (e.g. Gaussianity). This precludes the separation principle of classical LQG and standard Kalman filtering.
> > >
> > > 2. Data-rate Constraint:  Due to network limits, we must prioritize task-relevant features as opposed to equally weighting and sending the full $\hat{s}_t$, which a classic state observer in LQG would do. Specifically, in Line 125, $\phi_t$ has dimension $Z << n$ and we have rank constraints in Equations 9-10 for the linear setting.
> > >
> > > 3. Distinct Information Spaces: In classical tele-operation and networked LQG (Tatikonda ea 2004, Schenato ea 2007, Kostina ea 2019, etc) both states and controls would need to be compressed and sent across a bandwidth-limited link. In stark contrast, our formulation captures how local actuation can be improved by task-relevant forecasts of trends in $s_t$. Specifically, in line 125 and Eq 6, controls $\hat{u}_t$ are generated at the plant and not sent across the network like classical tele-operation.
> > >
> > > Moreover, unlike classical networked LQR, our network operator and controller measure *distinct, physically-separated information spaces for $s_t$ and $x_t$*. The operator only sees uncontrollable $s_t$ (e.g. cell demand) and not an application’s controllable state $x_t$ (e.g. taxi locations) nor controls $u_t$ (Line 125). Thus, *since only $x_t$ is local and controllable, while $s_t$ is external, it is inappropriate to club them as a controllable, macro-state $[x_t, s_t]$ like classical tele-operation.*

---

> > > > ### Comment · Reviewer_rL7L · 2021-08-27
> > > > **Answer to 2nd response**
> > > >
> > > > I appreciate the clarification of technical contributions and differences to related work.  Exactly such discussion (relating to prior work considering exogenous inputs) was missing in the original authors' response and in the paper.  IMO, such discussion (obviously shorter) should be in the paper, too.

---

> > > > > ### Author Response · Authors · 2021-08-28
> > > > > **Thank you  - we will definitely add this discussion.**
> > > > >
> > > > > Thanks for appreciating our response. Indeed, we will succinctly capture this clarification in the final version of the paper and cite Singh, Pal 2017.

---

### Decision · Program_Chairs · 2021-09-27

**Decision:**

Accept (Poster)

**Comment:**

This paper proposes an architecture and gradient-based training for an encoder/decoder scheme to reduce communication of an exogenous signal to a remote optimization-based controller. The proposed method takes the control objective into account when training the compression scheme, so that the encoder/decoder is optimized for the task at hand.

This paper introduces a (to some extend) novel problem, presents theoretical results for the LQR control special case and  demonstrates improved performance on a set of benchmark tasks. However, the reviewers also find that the proposed method is a mix and patch of different techniques (though the proposed combination seems novel) and the numerical demonstrations could be strengthened by including additional baselines.

This paper tackles an problem that might become increasingly important in the future. The reviewers therefore advocate the acceptance of the work, but assume that the criticized details will be addressed in the revision, including
- a discussion of the stochastic problem variation
- clarification of technical contributions and differences to related work (citing (Singh, Pal, 2017) is not the crucial point here, rather that the discussion is _specifc_  (as drafted in the author's response))